# A/B Testing for Recommender Systems in a Two-sided Marketplace

**Preetam Nandy, Divya Venugopalan, Chun Lo, Shaunak Chatterjee**
LinkedIn Corporation
Mountain View, CA 94083
{pnandy, dvenugopalan, chunlo, shchatterjee}@linkedin.com

## Abstract

Two-sided marketplaces are standard business models of many online platforms (e.g., Amazon, Facebook, LinkedIn), wherein the platforms have consumers, buyers or content viewers on one side and producers, sellers or content-creators on the other. Consumer side measurement of the impact of a treatment variant can be done via simple online A/B testing. *Producer side measurement is more challenging because the producer experience depends on the treatment assignment of the consumers*. Existing approaches for producer side measurement are either based on graph cluster-based randomization or on certain treatment propagation assumptions. The former approach results in low-powered experiments as the producer-consumer network density increases and the latter approach lacks a strict notion of error control. In this paper, we propose (i) a quantification of the quality of a producer side experiment design, and (ii) a new experiment design mechanism that generates high-quality experiments based on this quantification. Our approach, called UniCoRn (Unifying Counterfactual Rankings), provides explicit control over the quality of the experiment and its computation cost. Further, we prove that our experiment design is optimal to the proposed design quality measure. Our approach is agnostic to the density of the producer-consumer network and does not rely on any treatment propagation assumption. Moreover, unlike the existing approaches, we do not need to know the underlying network in advance, making this widely applicable to the industrial setting where the underlying network is unknown and challenging to predict a priori due to its dynamic nature. We use simulations to validate our approach and compare it against existing methods. We also deployed UniCoRn in an edge recommendation application that serves tens of millions of members and billions of edge recommendations daily.

## 1 Introduction

Learning via experiments is one of the most powerful and popular ways to improve in many domains of life. In the tech industry, experiments are very commonplace to better understand user preferences and how to serve them best. Such experiments, known as A/B testing or bucket tests [5, 6, 16, 19], are performed by randomized allocation of a treatment and control variant to some population and measuring the average treatment effect (ATE) [1, 4] relative to control. The populations receiving treatment and control are statistically identical since they were randomly selected.

A/B testing is a powerful tool because of its design simplicity and ease of setup. It is accurate in applications where the behavior of a measurement unit (e.g., a user) is unaffected by the treatment allocated to any other measurement unit. This principle is called "Stable Unit Treatment Value Assumption" or SUTVA [11–13]. The SUTVA principle is reasonably accurate for experiments in

several viewer side applications of recommender systems (e.g., newsfeed ranking, search), where each viewer acts independently based only on what is shown to her.

In marketplace settings, the SUTVA condition is often violated. A bipartite graph[1] is a common abstraction for two-sided marketplaces. Let us consider the example of content recommendation in a newsfeed ranking application with content viewers and producers. While the effect of a ranking change (e.g., showing more visual content) conforms to SUTVA on the viewer side, the effect on the producer side (i.e., the impact on producers who post more/less visual content) does not. A producer's experience is affected by the allocation of treatment to all her potential viewers. For instance, a producer who primarily posts images will get more exposure (which directly affects her behavior) as more viewers are allocated to the new treatment. Sellers and buyers are an identical analogue to producers and consumers. Violations of SUTVA, especially on the producer or seller side experience, are commonplace in many marketplace experiments [8, 17] and form an important area of study, especially as marketplaces gain greater prominence.

A popular approach in experiment design for marketplaces is to partition the graph into near separable clusters [10, 14, 18]. Then each cluster is considered an independent mini-graph, and randomized treatment allocation is done at the cluster level (i.e., all nodes in that cluster are allocated the same treatment). This works well in sparse graphs where many such clusters can be found without ignoring too many edges. A different approach [2, 7], relevant especially in the advertising world, "creates" multiple copies of the universe by splitting the limited resources (e.g., daily budget) of entities on one side of the graph (e.g., advertisers). This works when the ecosystem has a periodic reset.

Another recent approach [9] designs an experiment by identifying a modified version of the treatment, which is allocated to a proportion of a node's network to mimic the effect on the node that allocating the original treatment to the node's entire network would have had. This works well for denser networks but makes assumptions on how the treatment effect propagates. Many of these approaches require knowing the network structure *a priori*, and hence do not work for dynamic graphs.

In this work, we propose a novel and simple experiment design mechanism to generate high quality producer side experiments, where the quality is defined by a design inaccuracy measure that we introduce (see Definition 1). The mechanism also facilitates choosing a desired trade off between the experiment quality and the computational cost of running it. **Our experiment design is shown to be optimal with respect to the inaccuracy measure**. Our solution is applicable to any ranking system[2], which forms the viewer side application in most marketplace problems. *The key insight is that there is a unique viewer side ranking corresponding to each producer side treatment when all producers are allocated to that treatment. When producers are allocated to different treatment variants (e.g., to run treatment and control variants simultaneously), there are multiple possibly conflicting "counterfactual" rankings. By unifying the different counterfactual rankings (hence "UniCoRn") based on treatment allocation on the producer side, and careful handling of conflicts, we obtain a high quality experiment.* A recent work [3] is a specific instance of our generalized design, relying on small producer ramps, which minimize chances of conflict between the counterfactual rankings.

Our solution is designed to work at ramps of any size (higher ramps are often necessary for sufficient power). Furthermore, it improves upon the limitations of most prior approaches. It is agnostic to the density of the graph, does not depend on any assumptions on how the treatment effect propagates, and we do not need to know the structure of the graph *a priori*. One downside is the online computation cost of running an experiment using our design, and we provide a parameter to control this cost.

The key contributions of our work are as follows:

- An inaccuracy based metric to quantify the quality of an experiment and a novel producer side experiment design mechanism that unifies multiple counterfactual rankings.

- We prove the optimality of our experiment design, as well as bias and variance bounds.

- We show through extensive simulations how the method performs in various synthetic scenarios and against multiple existing approaches [3, 9].

- A real-world implementation of the proposal in an edge recommendation problem.

---

[1]This paper is focused on two-sided marketplaces. For more than 2 sides, a multi-partite graph can be used.
[2]Including single-slot and multi-slot ranking applications

The rest of the paper is structured as follows. Section 2 describes the problem setup in the context of a bipartite graph. The UniCoRn algorithm is presented in Section 3 along with an example demonstrating the different steps and certain theoretical properties of our method, including its optimality. In Section 4, we demonstrate the robustness of our method through detailed simulation studies, and we share our experience implementing UniCoRn in an edge recommendation application in one of the biggest social network platforms in the world. Finally, we conclude in Section 5 with a discussion of some extensions of our work and its general implications.

## 2   Problem setup

Let us consider a bipartite graph linking two types of entities - producers and consumers. A recommender system recommends an ordered set of items generated by the producers to each consumer, where items (e.g., connection recommendations, content recommendations or search recommendations) are ordered based on their estimated relevance in that consumer session[3]. We use the terminology "consumer (or producer) side experience" to refer to a measurable quantity associated with a consumer (or producer) that depends on the rank assigned by the recommendation system. One can get an unbiased estimate of the consumer-side impact (with respect to a metric, outcome or response of interest) by randomly exposing two disjoint groups of consumers to the treatment model and the control model respectively and measuring the average difference between the treatment group and the control group.

This classical A/B testing strategy does not work for measuring the producer side impact since that depends on the consumers' treatment assignment and should be ideally measured by allocating the same treatment to all the consumers connected to the producer in question. As illustrated in Figure 1a, satisfying this ideal condition simultaneously for all (or many) producers is not possible. For instance, $consumer$ 3 is connected to $producer$ 2 (in control) and $producer$ 3 (in treatment).

**Notation and terminology**: We consider an experimental design $\mathcal{D}$ with mutually exclusive sets of producers $P_0, \ldots, P_\mathcal{K}$ corresponding to treatments $T_0, \ldots, T_\mathcal{K}$ respectively. We refer to $T_0$ as control model (or recommender system) and all other $T_k$ as treatment model(s). The size of $P_k$ is determined by the ramp fraction (i.e., treatment assignment probability) of the corresponding models. Let $p_k, k = 1, \ldots, \mathcal{K}$ denote the ramp fractions satisfying $\sum_{k=0}^{\mathcal{K}} p_k = 1$. An online experiment typically spans over a time window, in which each consumer can have zero to more than one sessions and each producer can produce zero to more than one items. We denote the set of all sessions and the set of all items by $\mathcal{S}$ and $\mathcal{I}$ respectively. In each session $s$, the set of items under consideration is denoted by $\mathcal{I}_s$, which is a subset of $\mathcal{I}$. The counterfactual rank $R_k(i, \mathcal{I}_s)$ is the rank of item $i$ in consumer session $s$ with items $\mathcal{I}_s$ when $all$ items are ranked by treatment $T_k$. We denote the rank of item $i$ in the experimental design $\mathcal{D}$ by $R_\mathcal{D}(i, \mathcal{I}_s)$. We use the notation $i \in P_k$ to denote that item $i$ belongs to a producer in $P_k$. We reserve the use of the letters $k$, $i$ and $s$ for indexing a treatment variant, referring to an item and denoting a session.

**Design accuracy and cost**: An experimental design to accurately measure the producer side experience should also have a reasonable computational cost (hereafter just "cost") of running the experiment. As Section 3 will show, the accuracy and the cost are conflicting characteristics of our experimental design. Thus, having the flexibility to explicitly trade-off accuracy against cost is desirable. To this end, we provide a quantification of these characteristics in terms of counterfactual rankings. To define accuracy, we compare the design rankings $R_\mathcal{D}(i, \mathcal{I}_s)$ with *the ideal (but typically unrealizable) ranking* $R^*(i, \mathcal{I}_s)$ that equals $R_k(i, \mathcal{I}_s)$ if $i \in P_k$. An example is shown in Figure 1b.

**Definition 1.** *The inaccuracy of the experimental design $\mathcal{D}$ is given by*

$$Inaccuracy(\mathcal{D}) := \mathbb{E}\left(R_\mathcal{D}(i, \mathcal{I}_s) - R^*(i, \mathcal{I}_s)\right)^2, \ \ where \ R^*(i, \mathcal{I}_s) = \sum_{k=0}^{\mathcal{K}} R_k(i, \mathcal{I}_s)\, 1_{\{i \in P_k\}}.$$

When no treatments are being evaluated online, each $i \in \mathcal{I}_s$ is scored only by the control model $T_0$. Section 3 shows that each item might be scored multiple times using different treatment models in an experiment design. We quantify this computational expense as the cost of the design.

---

[3]A session encapsulates the context of the recommendation request such as a news feed visit (in case of content recommendations) or a search query (in the case of search recommendations).

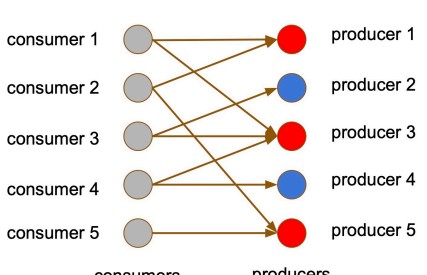

| | $P_0$ | $P_1$ |
|---|---|---|
| Item 1 | ✔ | ✗ |
| Item 2 | ✔ | ✗ |
| Item 3 | ✔ | ✗ |
| Item 4 | ✔ | ✗ |
| Item 5 | ✗ | ✔ |
| Item 6 | ✗ | ✔ |
| Item 7 | ✗ | ✔ |
| Item 8 | ✗ | ✔ |

(i) Problem setup

| | $R_0$ | $R_1$ | $R^*$ |
|---|---|---|---|
| Item 1 | 1 | 8 | 1 |
| Item 2 | 2 | 7 | 2 |
| Item 3 | 3 | 6 | 3 |
| Item 4 | 4 | 5 | 4 |
| Item 5 | 5 | 4 | 4 |
| Item 6 | 6 | 3 | 3 |
| Item 7 | 7 | 2 | 2 |
| Item 8 | 8 | 1 | 1 |

(ii) Unrealizable pair of counterfactual rankings

| | $R_0$ | $R_2$ | $R^*$ |
|---|---|---|---|
| Item 1 | 1 | 4 | 1 |
| Item 2 | 2 | 3 | 2 |
| Item 3 | 3 | 2 | 3 |
| Item 4 | 4 | 1 | 4 |
| Item 5 | 5 | 8 | 8 |
| Item 6 | 6 | 7 | 7 |
| Item 7 | 7 | 6 | 6 |
| Item 8 | 8 | 5 | 5 |

(iii) Realizable pair of counterfactual rankings

**(1a)** Bipartite graph of producers and consumers. Due to shared consumers between producers, it is infeasible to ensures all consumers connected to a producer get the same treatment as the producer.

**(1b)** Counterfactual Rankings. In (i), there are two mutually exclusive sets of producers $P_0$ and $P_1$, each of size 4. In (ii), giving each item their ideal position in a unified ranking is not possible since there are conflicts. In (iii), there are no such conflicts and the ideal unified counterfactual ranking is realizable.

**Definition 2.** *Let $N_{\mathcal{D}}(i, \mathcal{I}_s)$ denote the total number of times a scoring function (i.e. one of $T_k$'s) needs to be applied to to obtain a ranking of the items in $\mathcal{I}_s$ according to $\mathcal{D}$. The cost of an experimental design $\mathcal{D}$ is given by $Cost(\mathcal{D}) := \mathbb{E}\left(N_{\mathcal{D}}(i, \mathcal{I}_s)\right).$*

Now that we have an inaccuracy measure and a cost metric, we can define an experiment design algorithm that allows us to choose a desired balance between the two.

## 3 UniCoRn: Unifying Counterfactual Rankings

A typical recommender system comprises of a (possibly composite) model (machine learnt or otherwise) that assigns a relevance score to each candidate item. Items are then ranked according to their scores (higher the score, lower the rank; ties broken randomly). We want to measure the impact of a new recommender system ($T_1$) compared to the control recommender system ($T_0$) on producers (or sellers) in a two-sided marketplace via online A/B testing. Many recommender systems in industry have two phases: (i) a candidate generation phase, which considers a much larger set of candidates, followed by (ii) a ranking phase using a more sophisticated model with higher computation cost and hence often scoring much fewer items. Minor modifications needed to handle such multi-phase systems are covered in Section 4.3. Until then, we focus on single phase ranking systems. We also assume one treatment and one control for now, and extend to multiple treatments in Section 3.2.

### 3.1 The UniCoRn algorithm

For given disjoint producer sets $P_0$ and $P_1$ corresponding to $T_0$ and $T_1$ respectively, we present a class of experimental designs $UniCoRn(P_0, P_1, \alpha)$ parametrized by the tuning parameter $\alpha \in [0, 1]$ controlling the cost of the experiment. Recall that $\{R_k(i, \mathcal{I}')\}$ denotes a ranking of the items in $\mathcal{I}'$ according to $T_k$ in descending order (i.e. $T_k(i) \geq T_k(j)$ implies $R_k(i, \mathcal{I}') \leq R_k(j, \mathcal{I}')$) for $k = 0, 1$.

For each consumer session $s$, the $UniCoRn(P_0, P_1, \alpha)$ algorithm provides a ranking $\{R_{\mathcal{D}_U}(i, \mathcal{I}_s)\}$ of the items in $\mathcal{I}_s$ such that the rank of item $i \in P_k$ is close to $R_k(i, \mathcal{I}_s)$ simultaneously for all $i \in \mathcal{I}_s$ and $k = 0, 1$. *Please note that the underlying consumer-producer graph is not needed to apply $UniCoRn(P_0, P_1, \alpha)$.* The detailed steps of UniCoRn are provided in Algorithm 1 and Figure 2a provides a visual walkthrough of $UniCoRn$ using an example. The key components are:

- **Initial slot allocation** (Step 2): Identify positions allocated to all items using $T_0$.

- **Obtain mixing positions** (Steps 3 - 5): Identify the slots $\mathcal{L}$ to mix up and accommodate the two counterfactual rankings, and the slots that will not partake in this process.
  - $\alpha$ determines the fraction of $P_0$ items and slots that will be used in the mixing
  - All items in $P_1$ and their corresponding slots participate in the mixing

- **Perform mixing** (Steps 6 - 8): Obtain the *relative rank* of each item using the score according to that item's treatment assignment. Use these relative ranks to blend items (that were selected for mixing) from different groups with ties broken randomly (see Figure 2a).

**Algorithm 1** $UniCoRn(P_0, P_1, \alpha)$

---

**Require:** producer sets $P_0, P_1$, scoring models $T_0$ and $T_1$ and tuning parameter $\alpha$;
**Ensure:** a ranking of items for each session $s$;

1: **for** Each session $s$ with item set $\mathcal{I}_s$ **do**
2:     Get a ranking of all items $\{R_0(i, \mathcal{I}_s)\}$ according to $T_0$;
3:     Construct $P_0^*$ by randomly selecting producers from $P_0$ with probability $\alpha$;
4:     Let $\mathcal{I}_{s,0}$, $\mathcal{I}_{s,1}$ and $\mathcal{I}_{s,0}^*$ be the sets of items with producers in $P_0$, $P_1$ and $P_0^*$ respectively;
5:     Find the rank positions $\mathcal{L} = \{R_0(i, \mathcal{I}_s) : i \in \mathcal{I}_{s,1} \cup \mathcal{I}_{s,0}^*\}$ of the items in $\mathcal{I}_{s,1} \cup \mathcal{I}_{s,0}^*$;
6:     Obtain rankings $\{R_0(i, \mathcal{I}_{s,1} \cup \mathcal{I}_{s,0}^*)\}$ and $\{R_1(i, \mathcal{I}_{s,1} \cup \mathcal{I}_{s,0}^*)\}$ according to $T_0$ and $T_1$;
7:     Compute the following rank-based score

$$rank\_score(i) = R_0(i, \mathcal{I}_{s,1} \cup \mathcal{I}_{s,0}^*)\, 1_{\{i \in P_0^*\}} + R_1(i, \mathcal{I}_{s,1} \cup \mathcal{I}_{s,0}^*)\, 1_{\{i \in P_1\}};$$

8:     Rerank $\mathcal{I}_{s,1} \cup \mathcal{I}_{s,0}^*$ in the positions $\mathcal{L}$ based on $rank\_score(i)$ in ascending order (i.e., $rank\_score(i) \leq rank\_score(j)$ implies $rank(i) \leq rank(j)$) while breaking ties randomly;

---

In Algorithm 1, we guarantee that in the final ranking (i) the ordering among the items in $P_0$ respects the $T_0$ based ranking, (ii) the ordering among the items in $P_1$ respects the $T_1$ based ranking, and (iii) the distribution of the rank of a randomly chosen item in $P_0$ is the same as the distribution of rank of a randomly chosen item in $P_1$ (i.e., no cannibalization) and the common distribution is $Uniform\{1, ..., K\}$ if there are k slots. It is easy to see that (i) and (ii) hold by design and (iii) follows from the fact that $\mathcal{L}$ is a uniform sample from $\{1, ..., K\}$ as $P_0$ and $P_1$ are independent of the ranking distributions generated by $T_0$ and $T_1$ (due to randomized treatment allocation).

(2a) Algorithm 1 with $\alpha = 0.5$. (i) Complete rankings under $T_0$ and $T_1$. (ii) $P_0^*$ sampled as $\{Item\,2, Item\,3\}$. (iii) Separate ranking of $P_0^*$ using $R_0$ and $P_1$ using $R_1$. (iv) Unified ranking of $\mathcal{L} = P_0^* \cup P_1$ with all ties broken in favor of the orange items ($P_1$). (v) Final ranking obtained by placing items in $P_0 \setminus \mathcal{L}$ in their $R_0$ ranks, then remaining slots filled with the unified ranking of $\mathcal{L}$.

(2b) The impact of $\alpha$. $\alpha \in [0, 1]$ is an algorithm parameter that specifies the amount of flexibility in combining the two counter-factual rankings, with $\alpha = 0$ being the least flexible and $\alpha = 1$ being the most. As a result, $\alpha = 0$ incurs the highest inaccuracy but has the lowest cost, while $\alpha = 1$ is the most accurate and computationally expensive.

**The impact of** $\alpha$: We inserted $\alpha$ in our design to provide an explicit lever to control the balance between accuracy and cost. The more items (i.e., $|\mathcal{L}|$) we include in the mixing, the greater the accuracy. However, the mixing step requires every eligible item in $\mathcal{L}$ to be scored by every model, and hence the increased accuracy can come at a hefty cost. The implication of different choices of $\alpha$ in shown in Figure 2b, using the same example. All ties are broken in favor of items in $P_1$. Let $c_k$ denote the computation cost [4] of scoring all items (from all producers, that is) using $T_k$, where the scoring cost of each item is the same. Then the total cost is given by $c_0 + (\alpha p_0 + p_1)c_1$.

### 3.2 Handling multiple treatments

Thus far, we have considered one treatment and one control. Simultaneous measurement of multiple treatments (against a control variant) can be achieved with a simple extension to the mixing selection step. The effect of each treatment can be observed by independently comparing the corresponding treatment population to the control population. As a quick recap of critical notation, $T_0$ denotes the control model. With $\mathcal{K}$ treatments in total and $p_k$ denoting the ramp fraction of $T_k$, $\sum_{k=0}^{\mathcal{K}} p_k = 1$.

---

[4]Using Definition 2, $c_0 = c_1 = \mathcal{I}_s$. This refinement can handle variable model complexities.

**Greater mixing:** This is the trivial extension of Algorithm 1. We first fix positions of $1 - \alpha$ fraction of items from $P_0$ and then mix the remaining $P_0$ items with **all items from each** $P_k$ for $k = 1, \cdots, \mathcal{K}$. This family of designs (by varying $\alpha$) has higher cost and lower inaccuracy. Hence, it is suitable for offline scoring applications and online applications without strict scoring latency constraints. The total (computation) cost is given by $c_0 + (1 - (1 - \alpha)p_0) \sum_{k \geq 1} c_k$.

**Limited mixing:** An alternative is to select $\mathcal{L}$ by picking $\alpha$ **fraction of items from each** $P_k$ including $k = 0$. This reduces the cost in the mixing step since $\mathcal{L}$ is smaller and fewer items are scored by all models under consideration, but increases inaccuracy (compared to greater mixing) since lesser mixing happens. It is better suited for online applications with stricter latency requirements. The total computational cost is given by $c_0 + \alpha \sum_{k \geq 1} c_k + (1 - \alpha) \sum_{k \geq 1} p_k c_k$.

Next, we analyze some theoretical properties of this design in the two treatment scenario. At $\alpha = 1$, "greater" and "lesser" mixing scenarios are identical and the amount of mixing is the maximum possible. It is not surprising that this is also when the experiment design is provably optimal.

### 3.3 Theoretical results

We first prove the optimality of $UniCoRn(P_0, P_1, 1)$ with respect to the design inaccuracy measure given in Definition 1. Next, in Theorem 2, we provide bias and variance bounds for $UniCoRn(P_0, P_1, 1)$ and we show that our bounds are tight in the sense that the equality can be achieved in an adversarial situation. Proofs of all the results are given in the appendix.

**Theorem 1** (Optimality of $UniCoRn(P_0, P_1, 1)$). *Let $\mathcal{D}_U$ be a design based on Algorithm 1 with randomly chosen $P_0$ and $P_1$, and with $\alpha = 1$. Then for any other design $\mathcal{D}$*

$$\mathbb{E}\left(R_{\mathcal{D}_U}(i, \mathcal{I}_s) - R^*(i, \mathcal{I}_s) \mid \mathcal{I}_s\right)^2 \leq \mathbb{E}\left(R_{\mathcal{D}}(i, \mathcal{I}_s) - R^*(i, \mathcal{I}_s) \mid \mathcal{I}_s\right)^2, \tag{1}$$

*where $R^*$ is as in Definition 1 with $\mathcal{K} = 1$. Equation (1) implies the optimality of $\mathcal{D}_U$ with respect to the design inaccuracy measure given in Definition 1, i.e. $Inaccuracy(\mathcal{D}_U, T_0, T_1) \leq Inaccuracy(\mathcal{D}, T_0, T_1)$ for all $T_0, T_1$ and for all design $\mathcal{D}$. The same results hold for the multiple treatment case described in Section 3.2.*

**Theorem 2** (Bias and Variance Bounds). *Let $\mathcal{D}_U$ be a design based on Algorithm 1 with randomly chosen $P_0$ and $P_1$, and with $\alpha = 1$. Then, for $k \in \{0, 1\}$, the conditional bias and the conditional variance of the observed rank $R_{\mathcal{D}_U}(i, \mathcal{I}_s)$ given $\mathcal{A}_{s,k,i,r} = \{\mathcal{I}_s, R^*(i, \mathcal{I}_s) = r, i \in P_k\}$ is given by*

1. *$|\mathbb{E}\left(R_{\mathcal{D}_U}(i, \mathcal{I}_s) - R^*(i, \mathcal{I}_s) \mid \mathcal{A}_{s,k,i,r}\right)| \leq c(k, p_1)$, and*

2. *$\mathrm{Var}\left(R_{\mathcal{D}_U}(i, \mathcal{I}_s) \mid \mathcal{A}_{s,k,i,r}\right) \leq 2\min(r - 1, |\mathcal{I}_s| - r)\, p_1\, (1 - p_1) + c(k, p_1)\, (1 - c(k, p_1)),$*

*where $p$ is the probability of assigning an item to the treatment group $P_1$, $R^*$ is as in Definition 1 with $\mathcal{K} = 1$, and $c(k, p_1) = \{k(1 - p_1) + (1 - k)p_1\}/2$. The equality holds in both cases when $r \neq \frac{|\mathcal{I}_s|+1}{2}$ and the treatment ranking $\{R_1(i, \mathcal{I}_s)\}$ is the reverse of the control ranking $\{R_0(i, \mathcal{I}_s)\}$ with probability one.*

For $UniCoRn(P_0, P_1, \alpha)$ with $\alpha < 1$, for all $i \in P_0 \setminus P_0^*$, we have $R_{\mathcal{D}}(i, \mathcal{I}_s) = R^*(i, \mathcal{I}_s)$, implying zero bias and zero variance. For all $i \notin P_0 \setminus P_0^*$, it is easy to see that $R_{\mathcal{D}}(i, \mathcal{I}_s)$ can be written as

$$R_{\mathcal{D}}(i, \mathcal{I}_s) = X + (r - 1 - X) \times R_{\mathcal{D}}(i, \mathcal{I}_{s,1} \cup \mathcal{I}_{s,0}^*)$$

where $X$ has a $Binomial(r - 1, (1 - \alpha)(1 - p_1))$ distribution, and $X$ and $R_{\mathcal{D}}(i, \mathcal{I}_{s,1} \cup \mathcal{I}_{s,0}^*)$ are conditionally independent given the ordered set of items $D_{0,[|\mathcal{I}_s|]}$ according to $T_0$. Therefore, the bias and bounds can be derived using the results in Theorem 2. We leave detailed computations to the interested reader. Next, we empirically evaluate the impact of $\alpha$ on design inaccuracy and implement UniCoRn to evaluate the producer side impact of a large-scale recommender system.

## 4 Empirical Evaluation

We analyze various aspects of the design inaccuracy in Section 4.1, followed by an analysis of the treatment effect estimation error with specific rank to response functions in Section 4.2. We conclude this section by sharing our experience of implementing $UniCoRn$ in a large-scale edge recommendation application for one of the largest social networks with 750+ million members, demonstrating the scalability of our algorithm.

For Sections 4.1 and 4.2, we create a simulated environment with $L = 100$ positions to generate data for the empirical evaluation of $UniCoRn(P_0, P_1, \alpha)$ (in short, $UniCoRn(\alpha)$). First, we compare the design accuracy and the cost of the variants of $UniCoRn(\alpha)$ based on a number of values of $\alpha$. Next, we compare the performances of $UniCoRn(\alpha)$ for $\alpha \in \{0, 0.2, 1\}$, the counterfactual ranking method of [3] (we will refer to this as $HaThucEtAl$) and a modified version of $OASIS$ [9] for estimating the average treatment effect. To the best of our knowledge, these are the only existing methods that do not require the underlying network to be known a priori. We implemented[5] the Algorithms in R.

## 4.1 Impact of $\alpha$

For a fixed treatment proportion $TP = |P_1|/(|P_0| + |P_1|)$, the cost (Definition 2) of $UniCoRn(\alpha)$ increases with $\alpha$. We present the cost and inaccuracy results for different values of $TP$, while taking the average over random choices $P_0$ and $P_1$. We also consider four different simulation settings corresponding to different levels of correlation $\rho \in \{-1, -0.4, 0.2, 0.8\}$ between treatment and control scores for comparing the design accuracy. We generated the scores from a bivariate Gaussian distribution. More data generation details are in Appendix A.2.

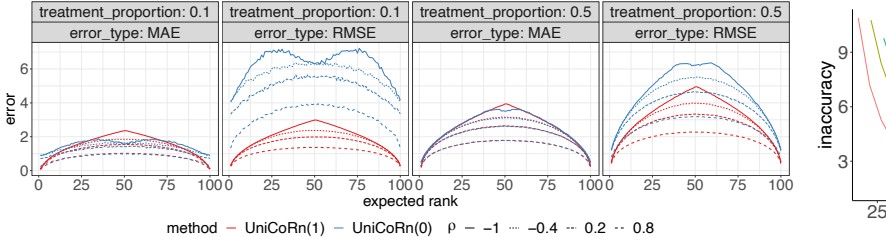

**(3a)** Average ranking errors for two measures of inaccuracy (MAE and RMSE) and for two different values of the treatment proportion (0.1 and 0.5) based on $N_S = 50000$ sessions with $L = 100$ slots each.

**(3b)** Cost vs. (in)accuracy trade-off at different treatment proportions (TP).

We consider two measures of inaccuracy (see Appendix A.2 for detailed definitions): (i) mean absolute error (MAE) and (ii) root mean squared error (RMSE). Figure 3a shows that the performance of $UniCoRn(0)$ and $UniCoRn(1)$ are roughly similar (or slightly better for $UniCoRn(0)$) with respect to MAE, but $UniCoRn(1)$ outperforms $UniCoRn(0)$ with respect to RMSE (validating Theorem 1). This is because $R_{\mathcal{D}}(i, \mathcal{I}_s) - R^*(i, \mathcal{I}_s) = 0$ for all items in $P_0$ for $UniCoRn(0)$, but the errors corresponding to the items in $P_1$ are much larger for $UniCoRn(0)$ compared to $UniCoRn(1)$. Note that the slightly better performance of $UniCoRn(0)$ with respect to MAE does not contradict the optimality result in Theorem 1, which is based on squared errors instead of absolute errors. Another interesting finding from Figure 3a is that a smaller value of $\rho$ (where -1 is the smallest value) corresponds to a more challenging design problem due to the increasing number of conflicts in the counterfactual rankings (cf. the last part of Theorem 2).

The cost (Definition 2) and inaccuracy (Definition 1) trade-off for a fixed value of $\rho = 0.8$ is shown in Figure 3b for different values of the treatment proportion $TP$. For each value of $TP$, we obtain the plot by varying $\alpha \in [0, 1]$. Since we directly generated the scores from a bivariate Gaussian distribution, the cost show in Figure 3b is a hypothetical cost according to Definition 2. As we see, designing an experiment with a higher $TP$ is more challenging than one with a lower $TP$ due to the increasing number of conflicts in the counterfactual rankings. Additionally, we see that experiments with a lower $TP$ are more sensitive to the choice of $\alpha$.

## 4.2 Comparison with existing methods

Note that $HaThucEtAl$ is designed for small ramp experiments. Following the authors' guidelines [3], we will be limiting ourselves to the case where 10% of the population is in control and 10% of the population is in treatment. For $UniCoRn(\alpha)$ and $OASIS$, we consider two different settings, namely (i) 10% treatment and 90% control and (ii) 50% treatment and 50% control.

---

[5]Code is available in the supplementary material.

OASIS solves a constrained optimization problem to match each producer's total counterfactual scores and a post-experiment adjustment corrects for mismatches. However, we consider a modification of OASIS which is a score-based counterpart of the rank-based $UniCoRn(1)$ algorithm. We assign a normalized counterfactual score to each item (i.e., no need for solving an optimization problem or for post-experiment correction). Following Section 5 of [9], we define normalized scores $p_k(s,i) = T_k(s,i) / \left( \sum_{i=1}^{L} T_k(s,i) \right)$, for $k = 0,1$. Then we define the counterfactual scores as $p^*(s,i) = \sum_{k \in \{0,1\}} p_k(s,i) 1_{\{i \in P_k\}}$.

We generate data from a simulated recommendation environment with $L = 100$ positions. Note that the computation cost shown in Figure 3a is hypothetical (based on Definition 2), as we generated the treatment and the control scores from (correlated) uniform distributions and hence we did not need to apply any scoring function. More data generation details are given in Appendix A.3. We consider the following two rank to response functions:

$$(avg\_fn)\ Y_i = \hat{E}\left[ \left( \frac{10}{\log(10+R_\mathcal{D}(i,\mathcal{I}_s))} \right)^2 \right] \text{ and } (max\_fn)\ Y_i = \max\left\{ \left( \frac{10}{\log(10+R_\mathcal{D}(i,\mathcal{I}_s))} \right)^2 \right\},$$

where the empirical average $\hat{E}$ and the max are over all items that appeared in a session and belong to producer $i$. We chose the logarithmic decay function $\frac{10}{\log(10+r)}$ to represent the value of a position (attention given to an item placed at position $r$) in a ranked list. Then we aggregate (using the average or the max function) the attention received by the items of a producer to define response functions. The treatment effects corresponding to $avg\_fn$ and $max\_fn$ are 0.16 and -0.87.

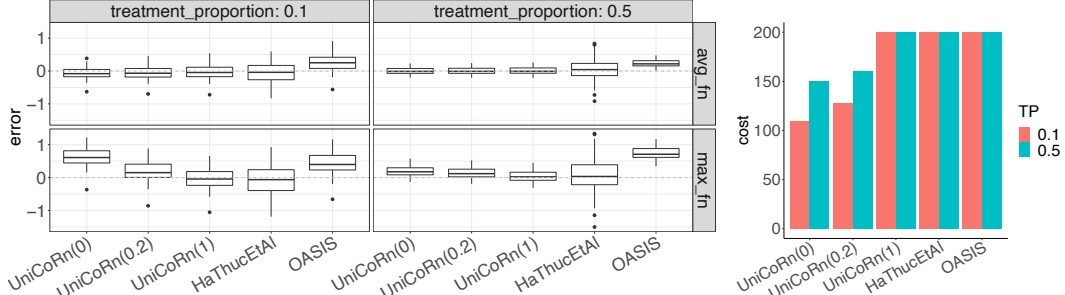

**(4a)** Errors in estimating the average treatment effect for two different rank to response functions and for two different values of the treatment proportion based on $N_S = 1000$ sessions with $L = 100$ slots each.

**(4b)** Hypothetical cost based on Definition 2 for treatment proportion (TP) equals 0.1 and 0.5.

Each iteration (based on 1000 sessions) including the data generation, reranking based on $UniCoRn(\alpha)$ for $\alpha \in \{0, 0.2, 1\}$, $HaThucEtAl$ and $OASIS$, and the treatment effect estimation took 36 seconds on average on a Macbook Pro with 2.4 GHz 8-Core Intel Core i9 processor and 32 GB 2667 MHz DDR4 memory. We repeat this 100 times and summarize the results in Figure 4a. Both $UniCoRn(\alpha)$ outperform OASIS (even for $\alpha = 0$) in terms of the treatment effect estimation error, demonstrating the advantage of rank-based methods over score-based methods. $UniCorn(1)$ and $Unicorn(0.2)$ outperform $HaThucEtAl$, as $HaThucEtAl$ exhibits a significantly higher variance due to its limitation to a 10% treatment and 10% control ramp. The performances of the variants of $UniCoRn(\alpha)$ are roughly equal for treatment proportion ($TP$) 0.5, whereas $UniCoRn(\alpha)$ is more sensitive to the choice of $\alpha$ at $TP = 0.1$. This is consistent with the findings in Figure 3b. Note that the sensitivity to the choice of $\alpha$ is more prominent when the rank to response function is $max\_fn$. This is consistent with Figure 3a, since $avg\_fn$ is a sub-linear function of the ranks, but $max\_fn$ is not. While accounting for the computational cost given in Figure 4b along with the estimation error in Figure 4a, the computationally cheapest method $UniCoRn(0)$ seems to be the best choice at $TP = 0.5$ whereas we need to choose the slightly more expensive variant $UniCoRn(0.2)$ to ensure an estimation quality as good as $UniCoRn(1)$.

### 4.3 Social Network application

Edge recommendations in social media platforms enable members to connect or follow other members. Edges also bring two sides of a marketplace together, e.g., content producers and consumers where content propagates along existing edges. Thus, edge recommendation products (see Figure 5

in the Appendix for a toy example) play a vital role in shaping the experience of both producers and consumers. The consumers of edge recommendations are "viewers" and A/B tests can measure the viewer side impact of any ranking change. The candidates (i.e., items) recommended are "viewees", because they are the members that are viewed and receive a connection request. Edge recommendations may have a large viewee impact, with number of viewees impacted often outnumbering viewers. To measure the viewee side effect, we implemented $UniCoRn$ in an online edge recommender system that serves tens of millions of members, and billions of edge recommendations daily. We chose $\alpha = 0$ (i.e., $UniCoRn(0)$) to minimize the online scoring latency increase. Next, we discuss two experiments conducted that cover candidate generation and scoring stage experiments. Key metrics include (i) Weekly Active Unique (WAU) users, i.e., number of unique users visiting in a week; and (ii) Sessions, i.e., number of user visits.

**Candidate generation experiment**: Large-scale recommender systems often have a candidate generation phase, which uses a simpler algorithm to evaluate a much larger set of items. The best few are then scored in the second ranking phase, which uses more sophisticated and computationally intensive algorithms. The two phases together comprise the ranking mechanism and $UniCoRn$ handles such scenarios with a simple extension. For any item $i$ selected by the control candidate selection model $C_0$ (or treatment $C_1$) but not by $C_1$ ($C_0$), the second phase scoring by treatment $T_1(i)$ (or control $T_0(i)$) is set to $-\infty$. The extension is detailed in the appendix (Section A.5).

In edge recommendation problems, a popular candidate generation heuristic is **number of shared edges**. This heuristic favors candidates with large networks. To neutralize this advantage, we tested a variant based on a **normalized version of shared edges** (i.e., fraction of the candidate's network that are shared edges with the viewer) and measured the impact using $UniCoRn(0)$. Thus, $C_0$ uses number of shared edges and $C_1$ uses the normalized version to generate candidates. The second phase ranking model was unchanged in this comparison, i.e., $M = T_0 = T_1$.

| Metrics | Delta % (candidate generation) | Delta % (ranking model) |
|---|---|---|
| Weekly Active Unique users | +0.51% | +0.13% |
| Sessions | +0.57% | +0.11% |

**Table 1:** Viewee side impact of a new candidate generation model (with the same ranking model as control) and a new ranking model (with the same candidate generation model as control), measured with 40% viewer side traffic. All results are highly significant with p-value < 0.001.

**Ranking model experiment**: The ranking stage scores all candidates based on the model assignment of the viewers. Ranking models may be composite models optimizing for viewer and/or viewee side outcomes. In one such experiment, the treatment model $T_1$ optimized for viewee side retention, i.e., we **boosted viewees likely to visit if they received an edge formation request**. Using $UniCoRn(0)$ and candidate set $I_s$, we obtain the ranking $\{R_0(i, \mathcal{I}_s)\}$ according to $T_0$ and find the positions $\mathcal{L} = \{R_0(i, \mathcal{I}_s) : i \in \mathcal{I}_{s,1}\}$. Then, we rescore candidates in positions $\mathcal{L}$ according to $T_1$ to obtain rankings $\{R_1(i, \mathcal{I}_{s,1})\}$ and rerank them within $\mathcal{L}$ to obtain the final list. $UniCoRn(0)$ is less costly because we rescore only the subset of candidates that belong to $P_1$.

**UniCoRn(0)'s implementation**: To generate the ranked list of viewees for a viewer, we first obtain the viewer treatment. If the viewer is not allocated to $UniCoRn$, we score all items using the allocated model (i.e., viewer treatment). This was also the flow prior to $UniCoRn$. If the viewer is allocated to $UniCoRn$, we then obtain the viewee treatment allocations for all viewees. The final ranking is obtained thus: (1) Score all items using a control model, (2) Obtain the viewee side treatment assignment for all viewees (i.e., items), (3) Score each viewee with the necessary treatments and blend using the scores (following Algorithm 1). The changes were implemented in Java in our distributed, real-time production serving system with no statistically significant serving latency added by this change.

**Results**: Table 1 shows viewee side results using 40% of the viewers[6] with $UniCoRn(0)$ for both the candidate generation and ranking change experiments. For each viewee $i$ (dest-member or producer), we compute the response $Y_i$ defined as the total count of the metric of interest in the experiment window (e.g., the number of visits in the experiment time window). The "Delta %" in Table 1 is the relative percentage difference between the average responses of the treatment and the control viewee

---

[6]We were unable to have 100% viewers in our experiment due to other parallel experiments, resulting in underestimation of the actual treatment effects corresponding to 100% viewer side ramps.

groups under the $UniCoRn(0)$ design. **Both experiments showed a positive change in WAUs and sessions** as they brought in more viewees onto the platform. Although the exact measurement fidelity could not be validated without the ground truth, we expected to observe a statistically significant positive impact. This is because we observed in a source-side experiment that the viewers tend to send invitations to more viewees under the treatment model than the control model, indicating a positive impact of the treatment model on the viewees. Note that these source-side measurements can be accurately obtained from a classical A/B testing setup on the viewer-side.

## 5 Discussion

A/B testing in social networks and two-sided marketplaces is extremely important to improve the experiences offered to various stakeholders. Our proposed experimentation design mechanism, $UniCoRn$, allows for high-quality producer side measurement with an explicit parameter to control the cost of the experiment at the expense of accuracy (or quality) loss in the measurement. Our experiment design is provably optimal, and our method has significant advantages over prior approaches: (i) It is agnostic to graph density (unlike, e.g., [15]), (ii) It makes no assumption on how the treatment effect propagates (unlike, e.g., [9]) or how the response depends on the treatment exposure (unlike, e.g., [10]), (iii) It lowers the variance of measurement (unlike, e.g., [3]), and (iii) It does not depend on knowing the graph structure a priori (unlike most existing methods, e.g., [9, 10, 15]).

**Limitations and Future work**: Our experiment design framework focuses on capturing the difference in exposure distribution of the producers in the treatment group and the producers in the control group. Hence, the $UniCoRn$ based treatment effect estimates would fail to capture some other types of differences between the treatment and the control. For example, a treatment may have an impact on a viewer's attention (e.g., the amount of time a viewer is spending on each session or the total number of viewer's sessions). This impact would not be captured by the $UniCoRn$ design, where all viewers receive a mix of treatment and control ranking.

The design accuracy measurement framework based on Definition 1 does not directly translate to the accuracy in the producer side treatment effect estimation without additional assumptions on the ranking to response function. We deliberately refrain from making such assumptions to build a more generally applicable experiment accuracy based framework. In the appendix, we discuss some additional assumptions under which the optimality result given in Theorem 1 can be extended to the treatment effect estimation problem. An interesting future direction could be to explore other types of loss functions in Definition 1 and study their connections with treatment effect estimation accuracy.

While $UniCoRn$ is designed to measure the producer side effect, sometimes the two sides of a marketplace are the same set of users playing different roles (e.g., a content producer is also a content consumer). In such scenarios, it may be important to measure the combined consumer and producer side effect. Such a measurement can be obtained by having a small set of producers, who are allocated to treatment, have their consumer experience ranked entirely based on that same treatment. This set has to be relatively small since the producer side experience will only be accurate if a large fraction of the consumers are on $UniCoRn$ (instead of pure treatment or pure control).

A related problem is to balance the power of the measurement (via higher producer side ramps) with the risk (which increases with larger consumer side ramps). Also, our proposed methodology can be extended to multi-partite graphs (i.e., marketplaces with more than two sides, such as food delivery platforms that connect users with drivers with restaurants). Such an extension would depend on the dynamics between the different graph partitions (i.e., entity types in the marketplace).

## 6 Acknowledgment

We would like to thank Parag Agarwal, Kinjal Basu, Peter Chng, Albert Cui, Weitao Duan, Akashnil Dutta, Aastha Jain, Aastha Nigam, Smriti Ramakrishnan, Ankan Saha, Rose Tan, Ye Tu and Yan Wang for their support and insightful feedback during the development of this system. We would also like to thank the anonymous reviewers for their helpful comments which has significantly improved the paper.

Finally, none of the authors received any third-party funding for this submission and there is no competing interest other than LinkedIn Corporation to which all authors are affiliated.

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
