$$= \frac{1}{|\mathcal{I}_s|} \sum_{i \in \mathcal{I}_s} \{ R_{\mathcal{D}}(i, \mathcal{I}_s) \}^2 + \frac{1}{|\mathcal{I}_s|} \sum_{i \in \mathcal{I}_s} \{ R^*(i, \mathcal{I}_s) \}^2 - \frac{1}{|\mathcal{I}_s|} \sum_{i \in \mathcal{I}_s} 2 \, R_{\mathcal{D}_U}(i, \mathcal{I}_s) \, R^*(i, \mathcal{I}_s). \quad (2)$$

The first term of (2) is identical for all $\mathcal{D}$ since $\{ R_{\mathcal{D}}(i, \mathcal{I}_s) \}$ is a permutation of $\{ 1, \dots, |\mathcal{I}_s| \}$, and the second term does not depend on $\mathcal{D}$. Therefore, it remains to show that

$$\sum_{i \in \mathcal{I}_s} R_{\mathcal{D}_U}(i, \mathcal{I}_s) \, R^*(i, \mathcal{I}_s) \geq \sum_{i \in \mathcal{I}_s} R_{\mathcal{D}}(i, \mathcal{I}_s) \, R^*(i, \mathcal{I}_s) \text{ for all } \mathcal{D}. \quad (3)$$

Without loss of generality, we assume $\mathcal{I}_s = \{ 1, \dots, |\mathcal{I}_s| \}$ and $R_{\mathcal{D}_U}(1, \mathcal{I}_s) \leq \cdots \leq R_{\mathcal{D}_U}(|\mathcal{I}_s|, \mathcal{I}_s)$. It is easy to see that for $\alpha = 1$, the $rank\_score(i)$ defined in Step 7 of Algorithm 1 equals $R^*(i, \mathcal{I}_s)$. This implies $\mathcal{D}_U$ ranks all items according to $\{ R^*(i, \mathcal{I}_s) \}$. Therefore, we must have $R_{1,s}^{T^*} \leq \cdots \leq R_{|\mathcal{I}_s|, s}^{T^*}$. Hence, the result in (3) follows from the rearrangement inequality, which states that

$$\sum_{i=1}^{n} x_{n+1-i} y_i \leq \sum_{i=1}^{n} x_{\sigma(i)} y_i \leq \sum_{i=1}^{n} x_i y_i$$

for every choice of real numbers $x_1 \leq \cdots \leq x_n$ and $y_1 \leq \cdots \leq y_n$, and for every permutation $x_{\sigma(1)}, \dots, x_{\sigma(n)}$.

Finally, $Inaccuracy(\mathcal{D}_U, T_0, T_1) \leq Inaccuracy(\mathcal{D}, T_0, T_1)$ follows directly from (1) by taking expectations over $s$, and all arguments given in this proof hold for the multiple treatment case (since we did not use $\mathcal{K} = 1$ in the proof).  $\square$

**Proof of Theorem 2.**  Fix an item $i$ with $R^*(i, \mathcal{I}_s) = r$. We only consider the case $r \leq (|\mathcal{I}_s| + 1)/2$ and $i \in P_1$. The results for the other cases follow from the same arguments with appropriate modifications.

For $k \in \{ 0, 1 \}$, we define $A_{r-1,k}$ to be the set of all items in $P_k$ that appears before item $i$ in the ranking according to $T_k$, i.e.,
$$A_{r-1,k} := \{ \ell : R_k(\ell, \mathcal{I}_s) \leq r - 1, \ \ell \in P_k \}.$$

Now it follows from the definition of $R_{\mathcal{D}_U}$ that $A_{r-1,k}$ is the set of all items in $P_k$ that appears before item $i$ in the observed ranking $\{ R_{\mathcal{D}_U}(\ell, \mathcal{I}_s) \}$.

Let
$$D_{k,[|\mathcal{I}_s|]} = (d_{k,1}, \dots, d_{k,|\mathcal{I}_s|})$$
denote the ordered set of items ranked according to $T_k$ for $k \in \{ 0, 1 \}$. Furthermore, let $Z_r$ be the indicator of the event that the ranking $\{ R^*(\ell, \mathcal{I}_s) \}$ contains two items, namely $d_{0,r}$ and $d_{1,r}$, with rank $r$:
$$Z_r = 1_{\{ d_{0,r} \in P_0, \ d_{1,r} \in P_1, \ d_{0,r} \neq d_{1,r} \}}.$$

Therefore,
$$R_{\mathcal{D}_U}(i, \mathcal{I}_s) = |A_{r-1,0} \cup A_{r-1,1}| + 1 + Z_r \times W, \quad (4)$$

where $W$ is an independent $Bernoulli(0.5)$ random variable. To see this, note that the first term counts the items that must come before item $i$ and second term counts item $i$ and the third term counts an additional item with probability $1/2$ whenever the ranking $\{ R^*(\ell, \mathcal{I}_s) \}$ contains two items with rank $r$.

Next, we derive the conditional distribution of the right hand side of Equation (4) given
$$\mathcal{E}_r = \{ D_{0,[r]}, \ D_{1,[r]}, \ \mathcal{I}_s, \ i \in P_k, \ R^*(i, \mathcal{I}_s) = r \}.$$

**Step 1 (Independence of $|A_{r-1,0} \cup A_{r-1,1}|$ and $Z_r$):**  Note that $|A_{r-1,0} \cup A_{r-1,1}|$ depends only on the treatment assignment of items in $D_{0,[r-1]} \cup D_{1,[r-1]}$, and $Z_r$ depends only on the treatment assignment of $\{ d_{0,r}, d_{1,r} \}$. Hence, $|A_{r-1,0} \cup A_{r-1,1}|$ and $Z_r$ are independently distributed given

$\mathcal{E}_r$.

**Step 2 (Distribution of $|A_{r-1,0} \cup A_{r-1,1}|$):** We decompose the first term of the right hand side of Equation (4) as follows:

$$|A_{r-1,0} \cup A_{r-1,1}| = Q_{r-1} + N_{0,r-1} + N_{1,r-1}, \tag{5}$$

where $Q_{r-1} = |D_{0,[r-1]} \cap D_{1,[r-1]}|$ and $N_{k,r-1}$ is the number of items in $D_{k,[r-1]} \setminus D_{1-k,[r-1]}$ that are in $P_k$ for $k = 0, 1$.

Since $(D_{0,[r-1]} \setminus D_{1,[r-1]})$ and $(D_{1,[r-1]} \setminus D_{0,[r-1]})$ are disjoint sets, by the same argument as in Step 1, $N_{0,r-1}$ and $N_{1,r-1}$ are independently distributed. Therefore, the conditional distribution of $|A_{r-1,0} \cup A_{r-1,1}|$ can be written as sum of two independent binomial distributions and a constant:

$$(|A_{r-1,0} \cup A_{r-1,1}|) \mid \mathcal{E}_r = Q_{r-1} + Binomial(r-1-Q_{r-1},\ p_1)$$
$$+ Binomial(r-1-Q_{r-1},\ 1-p_1).$$

**Step 3 (Distribution of $Z_r$):** By Definition 1, $R^*(i, \mathcal{I}_s) = r$ and $i \in P_1$ implies $R_1(i, \mathcal{I}_s) = r$. Therefore, $d_{1,r} = i$ and the conditional distribution of $Z_r$ given $\mathcal{E}_r$ is Bernoulli with probability

$$\mathbb{P}(Z_r = 1 \mid \mathcal{E}_r) = \mathbb{P}(d_{0,r} \in P_0) \times 1_{\{d_{0,r} \neq i\}} = (1 - p_1) \times 1_{\{d_{0,r} \neq i\}}.$$

By combining Steps 1-3 and Equation (4), we get

$$R_{\mathcal{D}_U}(i, \mathcal{I}_s) \mid \mathcal{E}_r = Q_{r-1} + B_1(r-1-Q_{r-1},\ p_1) + B_2(r-1-Q_{r-1},\ 1-p_1)$$
$$+ 1 + B_3(1,\ (1-p_1) \times 1_{\{d_{0,r} \neq i\}}) \times B_4(1,\ 0.5),$$

where $B_j(n_j,\ q_j)$'s are independently distributed binomial random variables with parameters $n_j$'s and $q_j$'s. Therefore,

$$\mathbb{E}\left(R_{\mathcal{D}_U}(i, \mathcal{I}_s) \mid \mathcal{I}_s,\ i \in P_k,\ R^*(i, \mathcal{I}_s) = r\right) = r + (1 - p_1) \times \mathbb{P}(d_{0,r} \neq i) \leq r + (1 - p_1) \quad \text{and}$$

and

$$\text{Var}\left(R_{\mathcal{D}_U}(i, \mathcal{I}_s) \mid \mathcal{I}_s,\ i \in P_k,\ R^*(i, \mathcal{I}_s) = r\right)$$
$$= \text{Var}\left(\mathbb{E}\left(R_{\mathcal{D}_U}(i, \mathcal{I}_s) \mid \mathcal{E}_r\right) \mid i \in P_k,\ R^*(i, \mathcal{I}_s) = r\right)$$
$$+ \mathbb{E}\left(\text{Var}\left(R_{\mathcal{D}_U}(i, \mathcal{I}_s) \mid \mathcal{E}_r\right) \mid i \in P_k,\ R^*(i, \mathcal{I}_s) = r\right)$$
$$= (1 - p_1)^2\ \mathbb{P}(d_{0,r} \neq i)\ [1 - \mathbb{P}(d_{0,r} \neq i)] + 2\ (r - 1 - \mathbb{E}(Q_{r-1}))\ p_1\ (1 - p_1)$$
$$+ \left\{ \frac{1 - p_1}{2} - \frac{(1 - p_1)^2}{4} \right\}\ \mathbb{P}(d_{0,r} \neq i)$$
$$\leq 2\ (r - 1)\ p_1\ (1 - p_1) + \frac{1 - p_1}{2} \left\{ 1 - \frac{(1 - p_1)}{2} \right\}.$$

Finally, note that $r \neq \frac{\mathcal{I}_s + 1}{2}$ and the treatment ranking $\{R_1(i, \mathcal{I}_s)\}$ is the reverse of the control ranking $\{R_0(i, \mathcal{I}_s)\}$ with probability one implies $\mathbb{P}(d_{0,r} \neq i) = 1$ and $\mathbb{E}(Q_{r-1}) = 0$, and hence the equality holds in both cases. This completes the proof. $\square$

## A.2  Data generation for Section 4.1

We consider a recommendation environment with $L = 100$ positions. For each session $s$, we generate $L$ i.i.d. control and treatment score pairs $\{(T_0(s, i),\ T_1(s, i)) : i = 1, \ldots, L\}$ from a bivariate Gaussian distribution with zero means, unit variances and correlation coefficient $\rho$. We consider multiple $\rho$ values, namely $\rho \in \{-1, -0.4, -0.2, 0.8\}$, to evaluate the effect of the (rank) correlation between the control and the treatment score on the design accuracy. We use the following two measures of inaccuracy:

1. Mean Absolute Error (MAE): $\hat{E}\left(|R_{\mathcal{D}}(i, \mathcal{I}_s) - R^*(i, \mathcal{I}_s)| \mid R^*(i, \mathcal{I}_s) = \ell\right)$ and

2. Root Mean Squared Error (RMSE): $\sqrt{\hat{E}\left(R_{\mathcal{D}}(i, \mathcal{I}_s) - R^*(i, \mathcal{I}_s) \mid R^*(i, \mathcal{I}_s) = \ell\right)^2}$,

where $R^*(i, \mathcal{I}_s)$ is the counterfactual ranking of item $i$ in session $s$, and $\hat{E}$ denotes the empirical average taken over $N_S = 50000$ sessions.

### A.3 Data generation for Section 4.2

To create a simulation environment, we generate $N_P = 1000$ producers with "quality" generated from a $Beta(2, 5)$ distribution. We consider a recommendation environment with $L = 100$ positions. For each session $s$, we randomly choose $L$ producers with replacement and for each chosen producer $i$ with quality $q(i)$ we generate the control and treatment scores from $Uniform[q(i), 1 + q(i)]$ and $Uniform[q(i), 2 \times q(i)]$ distributions respectively. We generate data corresponding to $N_S = 1000$ i.i.d. sessions.

### A.4 Edge Recommendation Product Example

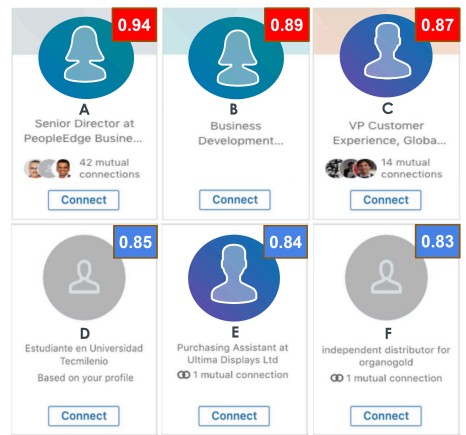
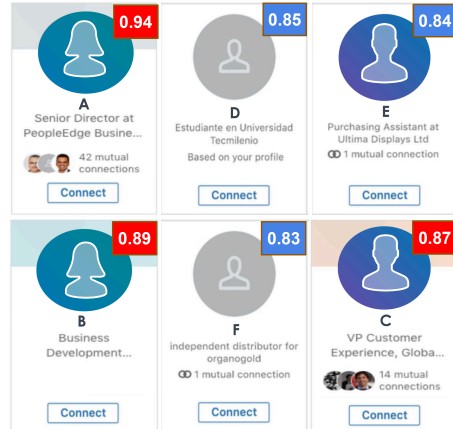

(a) Ranking Before UniCoRn      (b) Ranking After UniCoRn

**Figure 5:** A toy example for demonstrating UniCoRn based reranking in a sample edge recommendation product. Ranking of candidates with scores on top right. Red indicating treatment candidates with treatment model scores and blue the control candidates with control model scores. Sub-figure (a) shows ranking list without $UniCoRn$ whereas (b) shows ranking list with $UniCoRn$.

Figure 5 shows ranking without and with $UniCoRn$. $P_0$ and $P_1$ are exclusive and equally sized sets (i.e., with 50-50 split)[7]. Candidates A, B and C (scores in red) are in $P_1$ and scored by $T_1$. D, E, F (scores in blue) are in $P_0$ and scored by $T_0$. As $T_1$ has an additional boost, the scores for $P_1$ are typically higher than those of $P_0$ and would gain an unfair ranking advantage if combined without $UniCoRn$ (as shown in Figure 5(a)). $UniCoRn$ balances exposure to $P_0$ and $P_1$ (Figure 5(b)).

### A.5 Extension of $UniCoRn$ to a combination of candidate generation model and a ranking model

---
**Algorithm 2** $UniCoRnCandidateGeneration(P_0, P_1, \alpha)$

---
**Require:** producer sets $P_0$, $P_1$, candidate generation models $C_0$ and $C_1$, scoring models $T_0$ and $T_1$ and tuning parameter $\alpha$;
**Ensure:** a set of ordered items for each session $s$;
1: **for** Each session $s$ **do**
2:      Generate two sets of candidate items $\mathcal{I}_{s,C_0}$ and $\mathcal{I}_{s,C_1}$ based on $C_0$ and $C_1$ respectively;
3:      For $k = 0, 1$, define
$$T_{C_k}(i) = \begin{cases} T_k(i) & \text{if } i \in \mathcal{I}_{C_k} \\ -\infty & \text{Otherwise}; \end{cases}$$
4:      Order the items in $\mathcal{I}_s := \mathcal{I}_{s,C_0} \cup \mathcal{I}_{s,C_1}$ based on $UniCoRn(P_0, P_1, \alpha)$ with scoring models $T_{C_0}$ and $T_{C_1}$;

---

[7]Actual reference to the platform on which we implemented is hidden in the Figure to preserve anonymity during the review process. We will add back such information in the final submission.

## A.6 Discussion on the Optimality of $UniCoRn$

The optimality result presented in Theorem 1 does not guarantee the optimality of $UniCoRn$ to the average treatment effect estimation inaccuracy ($ATE\_inaccuracy$), except for some special cases. For example, the optimality result extends to $ATE\_inaccuracy(\mathcal{D})$ if $ATE\_inaccuracy = f(Inaccuracy)$ for a monotonic function $f(\cdot)$. To see this, note that $Inaccuracy(\mathcal{D}_u) \leq Inaccuracy(\mathcal{D})$ implies $f(Inaccuracy(\mathcal{D}_U)) \leq f(Inaccuracy(\mathcal{D}))$ for any monotonic function $f(\cdot)$. Although we cannot guarantee the optimality of $UniCoRn$ in a non-monotonic case, we can provide bounds on the $ATE\_inaccuracy$ in terms of $Inaccuracy$ for a class of smooth functions. For example, if $ATE\_inaccuracy = f(Inaccuracy)$ for a Lipschitz continuous function $f(\cdot)$ with $f(0) = 0$, then $|ATE\_inaccuracy| \leq c \times |Inaccuracy|$ where $c$ is the (smallest) Lipschitz constant for $f(\cdot)$. The results follows from the definition of Lipschitz continuity, i.e., $|f(x) - f(y)| \leq c \times |x - y|$ with $x = $ Design-inaccuracy and $y = 0$.

For another example, suppose the expected response of the i-th a producer equals $Y(i) = \sum_s g_s(R_{i,s})$ where $g_s(\cdot)$ is a session-specific monotonic function, then we can show optimality of $UniCoRn$ with respect to ATE-inaccuracy. To see this, note that the rearrangement inequality used in Eq. (3) in the proof of Theorem 1 would also hold for $g_s(R_{i,s})$'s and $g_s(R_{i,s}^*)$'s when $g_s(\cdot)$ is a monotonic function. An interpretation of $g_s(r)$ can be the attention given by a viewer in session $s$ at the position $r$ in the ranked list of items, which is often a monotonically decreasing function in most real-world recommendation systems. If $Y(i)$ is a nonlinear function of the $g_s(R_{i,s})$'s, then the optimality of $UniCoRn$ might not hold and the optimal design might depend on the functional form. We study this nonlinear case (with a max(.) function) in one of our simulation settings in Section 4.2 with $g_s(R_{i,s}) = [10/log(10 + R_{i,s})]^2$, and observed reasonably good (and better than existing methods) performance of $UniCoRn$.