# OpenReview forum: "A/B Testing for Recommender Systems in a Two-sided Marketplace"
_NeurIPS.cc/2021/Conference — NeurIPS 2021 Poster_

### Official Review · Reviewer_q14B · 2021-07-16

**Rating:** 6
**Confidence:** 4

**Summary:**

This paper studies the problem of conducting proper producer side A/B testing on 2-sided market/social networks. To be specific, the authors provide a series of heuristic-based methods to solve the ranking conflict which is common in producer side a/b testing because a users recommendation result can be filled with producers exploiting different ranking algorithms. The authors introduce a cost hyper-param to control the trade off between cost and accuracy. The authors provide both offline experiments and online use case study.


**Limitations And Societal Impact:**

I have some concerns and questions regarding this paper:

1.The method proposed by this paper is still a compromise. Because conflict resolving is needed, the ranking result in the treatment arm will be different compared to when the recommendation algorithm is fully launched to all producers.

A naive way to solve this is to create experiment arms based on dis-joint user cohort and apply different recommendation algorithms to all the producers within each arm. Then observe the impact of different recommendation algorithms on producers by comparing the aggregated producer-side performance from different arms.

Why is the proposed method better than the naive way? Naive way seems much cleaner.

2. The problem set up section can be further refined to make it easier for others to follow the objective of this paper. Current version is still a little bit difficult to follow.


**Main Review:**

The authors propose a very interesting problem, how to conduct proper producer side A/B testing and resolve potential ranking conflicts. This is interesting because most of A/B testing conducted in recommendation systems are consumer side A/B test and because of the disjoint of consumer-grouping, it would not face a lot of conflicting issues as mentioned in the paper.

But producer side A/B testing is very important as a lot of recommendation changes impact how producer's items(e.g. news stories, products) are consumed by users and will create a long term impact on recommendation systems. Methods proposed by this paper, though heuristic-based, can be beneficial for others to improve producer-side A/B test design.

The problem set up is actually quite complicated, but the authors provide a lot of intuition explanation and toy examples which help to make it easier for readers to follow and make the paper itself clearer.

I am also glad to see, in the evaluation part, the authors provide some use-case study on how this evaluation mechanism is deployed in some large scale recommendation applications, which makes the evaluation more convincing.


**Time Spent Reviewing:**

5

---

> ### Author Response · Authors · 2021-08-10
> **Response to Reviewer q14B**
>
> The naive way you mentioned might be the same as the cluster-based methods described in lines 49-53. We need to apply graph clustering here since the recommendation algorithms are not directly applicable to producers, but they have an indirect impact on the producers through their consumers. Therefore, we need to make sure that all consumers of a producer in treatment (or control) get the treatment model (or the control model). This can lead to conflicts as a producer in treatment, and a producer in control can share the same consumer (see Figure 1). Also, note that the consumer-producer graph (which may be unknown) is determined by the underlying environment (i.e., the existing market), and we cannot manipulate the graph during the experiment to create disjoint cohorts as in this case, we would be testing the treatment and the control model in a different environment (instead of the existing environment).
>
> As we mentioned in the paper, the cluster-based approaches do not work well for non-sparse graphs. Furthermore, one of the main advantages of our algorithm is that it does not require knowing the underlying graph (which can be dynamic in the experiment window or a priori unknown). To the best of our knowledge, it is not possible to apply the clustering-based approaches in the unknown and dynamic graph case. Although we did not directly compare our methods with cluster-based algorithms in simulations, we compare our methods with the OASIS algorithm of [Nandy et al., 2020], where the authors showed that OASIS performs much better than clustering-based algorithms in most realistic situations where the underlying graph cannot be separated into disjoint (and statistically identical) clusters.

---

### Official Review · Reviewer_7Bcd · 2021-07-17

**Rating:** 8
**Confidence:** 3

**Summary:**

In this paper, the author studied the problem of doing A/B testing for "producer" in two-side marketplace. Specifically, the author proposed a
 new metric to measure the inaccuracy in measuring the treatment effect for producers. The author also proposed a new experiment design and proves it to be optimal to the new metric. The experiment design was applied to a real world recommendation system.

**Limitations And Societal Impact:**

This paper would lead to large societal impact but I don't think it is a negative one.

**Main Review:**

Overall, the paper is well written. The problem is very realistic and has huge real world impact. The ideas are novel and the reasoning process is also very clear. I would like to see it being accepted.

Reasons to accept:
1. The paper is well written.
2. The problem has huge real world impact.
3. The ideas are novel, the author proposed a new metric to measure the in-accuracy of producer-side experiment, and it also proposed a new experiment design which is proved to be optimal to this measure.
4. The experiment design was applied to real world application.

Reasons to reject:
1. some minor issues about writing. For example, Line 16 "we prove that our experiment design is optimal". This seems to be over-claimed. It should be "we prove that our experiment design is optimal to the proposed in-accuracy measure"

**Time Spent Reviewing:**

2

---

> ### Author Response · Authors · 2021-08-10
> **Response to Reviewer 7Bcd**
>
> We are glad you see you appreciate the value of our work. It would be great if you could champion the paper in the discussion period.
>
> Your point on over-claiming the theoretical result is well taken. We’ll correct the claim you mentioned and double-check the others.

---

> > ### Comment · Reviewer_7Bcd · 2021-09-10
> > **Response to the authors**
> >
> > Thanks for the response.

---

### Official Review · Reviewer_aZdN · 2021-07-21

**Rating:** 5
**Confidence:** 4

**Summary:**

The authors propose a A/B testing framework that measures producer effectiveness in a two-sided setting.  The authors argue that while there are many ways to measure the consumer-side experience consistently, there is none on the producer-side that measures the effectiveness of a producer consistently, i.e., measure the experience of all consumers that use a particular producer.

**Ethical Concerns:**

None.

**Limitations And Societal Impact:**

None.

**Main Review:**

The authors contend that existing approaches to measure producer-side impact such as clustering-based randomizations produce 'low-powered experiments'.  It will greatly help if they can compare the outcomes of the UniCoRn algorithm with the clustering approach.  It is not clear why the set up is considered two-sided since impact of both sides is not measured simultaneously.

The paper is slightly hard to read.  How are the counterfactual rankings produced?  There can be a lot of them. In Fig. 2.iii, where is R_2 coming from and how are the ranks assigned?  There is a lot of work on rank aggregation (ranking producers under different user preferences and combining the orderings by minimizing the number of inversions under any individual ordering).  It'll help to compare the techniques used in this work to rank aggregation.

======

I read the authors' rebuttal and thank them for clarifying some of my misunderstanding.  I however feel that without addressing the impact on both sides of the market simultaneously, this is not a two-sided problem.  Also, it's not clear to me why the clustering approach cannot be used for a dynamic graph setting.  Naively, can't it could be done on an instance of the graph in each round of allocation?

======

**Time Spent Reviewing:**

2

---

> ### Author Response · Authors · 2021-08-10
> **Response to Reviewer aZdN**
>
> 1. There seems to be a significant misunderstanding on the objective of the paper. Our goal is NOT to measure “the effectiveness of a producer consistently, i.e., measure the experience of all consumers that use a particular producer.” Instead, we want to measure the impact of a new model (changing the ordering of the items shown to the consumers) on the producers. More precisely, we want to estimate the difference in the average response of the producers when all consumers are treated with model $B$ vs. the average response of the producer when all consumers are treated with model $A$.
>
> 2. For our empirical evaluation, we chose the OASIS algorithm of [Nandy et al., 2020] as one of the baselines instead of clustering-based algorithms. This is because OASIS is shown to outperform [Nandy et al., 2020] clustering-based algorithms in most realistic situations where the underlying graph cannot be separated into disjoint (and statistically identical) clusters. Furthermore, one of the main advantages of our algorithm is that it does not require knowing the underlying graph (which can be dynamic in the experiment window). To the best of our knowledge, it is not possible to apply the clustering-based approaches in the unknown and dynamic graph case.
> Measuring the producer-side impact in a two-sided marketplace is common terminology used by several other authors in this context. Usually, measuring the consumer-side impact is often a much easier task for which the standard A/B testing can be used. The simultaneous measurement is an important problem when the same set of users can play the role of consumers and producers. As we briefly discussed in Section 6, simultaneous measurement is possible with a simple extension of our setup.
>
> 3. We appreciate your feedback on the readability of the paper. If you could identify specific sections which were more difficult to follow, we will definitely look into improving them. Similar pointed feedback from another reviewer was quite helpful.
> For each treatment, there is a unique counter-factual ranking given by the ordering of the items generated by the treatment model. We guess you wanted to refer to Figure 1b.iii since there is no $R_2$ in any other figure. $R_i$ is an example (unique) ranking generated by a treatment model, say $T_i$, for $i$ = 1, 2, 3. In Figure 1b.iii, we give an example of rankings (generated by $T_0$ and $T_2$) that can be combined without any conflicts (unlike Figure 1b.ii, where we do get some conflicts for combining rankings generated by $T_0$ and $T_1$).
>
> 4. Thank you for pointing us to the rank aggregation literature. It might be an interesting future direction to change our current rank aggregation function in UniCoRn to achieve optimality with respect to another measure of inaccuracy, but these nontrivial extensions are out of the scope of this paper.

---

> > ### Author Response · Authors · 2021-08-25
> > **Response to Reviewer aZdN (Round 2)**
> >
> > Thank you for your active participation in the discussion. We would like to reiterate that simultaneous measurement is possible with a simple extension of our setup, and we have already discussed this in Section 5 (lines 362 - 368).
> >
> > In the clustering approach, each cluster gets randomly allocated to treatment or control. Now, if the cluster definitions are changing dynamically, it wouldn’t be possible to ensure that each producer receives the same treatment allocation every time (and without this, we cannot have the correct treatment-level experience of a producer). Moreover, it is computationally infeasible to run a clustering algorithm after every change in the graph in a large-scale social network with billions of members.

---

### Official Review · Reviewer_neLM · 2021-07-23

**Rating:** 7
**Confidence:** 4

**Summary:**

This paper introduces unicorn, a method to estimate the effect of different recommender systems on the producers in a platform. This is more challenging than consumer-side A/B testing where a simple randomized trial suffices since a producer's content will be recommended across multiple different treatments.
Unicorn works by ensuring the overall ranking of items presented to a consumer approximates the ranking of all treatments under consideration. The authors first present the algorithm for the case where there exists one control and one treatment, highlighting a parameter that allows them to tradeoff accuracy and cost. They then propose generalizations of unicorn to multiple treatments.
The paper goes on to present theoretical results bounding the bias and variance of unicorn (for one control/treatment with alpha set to 1), and showing their experimental design is optimal wrt the ranking error metric they defined.
Finally, they evaluate unicorn in simulation, comparing it to prior work. They also deploy unicorn in production on a large social media network.

**Limitations And Societal Impact:**

A/B testing and recommender systems are both subfields that have a very large impact on hundreds of millions of users. As such, I feel it is necessary that the authors address the potential societal impacts of their work. As it stands there is no such discussion.

I think the authors did a good job of addressing the technical limitations of their work in the discussion section.

**Main Review:**

Overall I think this is a well-written paper that addresses an important and novel problem that has seen too little attention from the research community. However, I do have some concerns I hope the authors can address. I have listed them roughly from most important to least important.
 - It seems the paper primarily focuses on minimizing the inaccuracy in Definition 1, both in their experiments and in the theoretical portion of their work. However, this is simply a proxy metric, what we hope to eventually measure is the average treatment effect, as the authors mention in the introduction. As a result, I was surprised that this seemed to not be considered in most of the paper. In the discussion section, the authors mention that their theoretical results extend to treatment effect estimation errors that are monotonic in their defined error metric, however, they show no proof of this, nor is it made clear whether common treatment effect estimates will be monotonic.
 - Following on the previous point, the simulated experiments in section 4.1 only measure MAE/RMSE with respect to the ideal ranking. Yet this would have been the ideal setting to measure the accuracy of various realistic treatment effect estimators under the unicorn framework. This would give the reader significantly more information regarding the reliability of the results they would obtain from running unicorn.
 - The motivation for the choice of rank to response functions defined on line 274 was unclear to me for the experiments in section 4.2.
 - I had a hard time interpreting the results presented in Table 1, how were the deltas computed? The caption mentions the results were highly significant with a very low p-value, however I couldn't find an explanation regarding how that p value was computed.
 - I wasn't sure what the takeaway should be for section 4.3,
 - I think the clarity of the paper could be improved by discussing the intuition behind why the algorithm work in section 3.1 alongside the already very well explained explanation as to how it works.

Given my concerns, I feel the paper might not yet be publication-ready. However, if the authors address my concerns (especially the top one regarding treatment effects), I am more than happy to raise my score.

**Time Spent Reviewing:**

5

---

> ### Author Response · Authors · 2021-08-10
> **Response to Reviewer neLM**
>
>
> We thank you for your thorough and constructive review. Please find our response to your questions below.
>
> 1. The optimal design for the average treatment effect measurement error (ATE-inaccuracy) would be a non-trivial extension of our current work and would probably require some untestable assumptions. However, our theoretical results and simulations do cover some interesting cases. First, as we claimed in the paper, the optimality of UniCoRn w.r.t. ATE-inaccuracy holds when it is a monotonic function of design-inaccuracy. To see this, note that design-inaccuracy($UniCoRn$) $\\leq$ design-inaccuracy(any other design) implies $f$(design-inaccuracy($UniCoRn$)) $\\leq$ $f$(design-inaccuracy(any other design)) for any monotonic function $f(.)$. Second, suppose the expected response of the $i$-th a producer equals $response(i) = \sum_s g_s(R_{i,s})$ where $g_s(.)$ is a session-specific monotonic function, then we can show optimality of UniCoRn w.r.t. ATE-inaccuracy. To see this, note that the rearrangement inequality used in Eq (3) in the proof of Theorem 1 in the appendix would also hold for $g_s(R_{i,s})$’s and $g_s(R^*_{i,s})$’s when $g_s(.)$ is a monotonic function. An interpretation of $g_s(r)$ can be the attention given by a viewer in session s at the position r in the ranked list of items, which is often a monotonically decreasing function in most real-world recommendation systems. If $response(i)$ is a nonlinear function of the $g_s(R_{i,s})$’s, then the optimality of $UniCoRn$ might not hold and the optimal design might depend on the functional form. We study this nonlinear case (with a max(.) function) in one of our simulation settings in Section 4.2 with $g_s(R_{i,s}) = [10 /log(10 + R_{i,s})]^2$, and observed reasonably good (and better than existing methods) performance of UniCoRn.
>
> 2. Based on the rationale above, we thought it was more prudent to decouple and separately analyze various aspects of the design inaccuracy first, followed by an analysis of the estimation error with specific rank to response functions. Hence, in Section 4.1, we validated our theoretical results based on ranking-inaccuracy, provided additional insights on the sensitivity of $\alpha$ (in $UniCoRn(\alpha)$) for RMSE and MAE, and demonstrated the cost-accuracy trade-off..
>
> 3. The logarithmic decay function $\frac{1}{\log(1+r)}$ is a popular choice in representing the value of a position (attention given to an item placed at position $r$) in a ranked list, as in the Normalized Discounted Cumulative Gain (NDCG) metric. Then we aggregate the attention received by the items of a producer to define a response function. We demonstrate how the sensitivity of $\alpha$ can depend on the choice of the aggregation function.
>
> 4. We ran the experiment for a week. For each viewee $i$ (dest-member or producer), we compute the response $Y_i$ defined as the total count of the metric of interest in the experiment window (e.g., # of visits in the experiment week). Then delta% = 100 * (the average response in the viewers in treatment - the average response in the viewers in control) / the average response in the viewers in treatment. The p-values are computed using the asymptotic Gaussian distribution (which follows from the central limit theorem and Slutsky’s theorem) of delta%.
>
> 5. In Section 4.3, we demonstrate the scalability of $UniCoRn$ with a large-scale recommendation application. We hope that this would provide practitioners with more confidence in deploying $UniCoRn$ in real-world applications.
>
> 6. Thanks for asking. We will add an expanded version of the following intuition. In Algorithm 1, we guarantee that in the final ranking (i) the ordering among the items in $P_0$ respects $T_0$, (ii) the ordering among the items in $P_1$ respects $T_1$, and (iii) the distribution of the rank of a randomly chosen item in $P_0$ is the same as the distribution of rank of a randomly chosen item in $P_1$ (i.e., no cannibalization) and the common distribution is $Uniform\\{1,..., K\\}$ if there are k slots. It is easy to see that (i) and (ii) hold by design and (iii) follows from the fact that $\mathcal{L}$ is a uniform sample from $\{1,..., K\}$ as $P_0$ and $P_1$ are independent of the ranking distributions generated by $T_0$ and $T_1$ (i.e., randomized treatment allocation).

---

> > ### Comment · Reviewer_neLM · 2021-08-20
> > **A few more questions**
> >
> > Thank you for your detailed response. I have a few follow-up questions. I have numbered them based on which point in your response I'm referring to.
> >
> > 1. (1) In that case, are you saying that all bets are off when the ATE-inaccuracy is not monotonic in terms of design accuracy? I don't think that's a fatal flaw, but I think this limitation should be clarified early in the paper. In particular, I think it would be really helpful if the paper briefly mentioned some common scenarios where an assumption of monotonicity is/isn't reasonable.
> > 2. (1) Would you be willing to include the exposition/proof you just wrote in the paper? I think this would significantly strengthen the paper by guiding practitioners in knowing when UniCoRn is/isn't applicable to their problem and what errors they can expect wrt their ATE estimates.
> > 3. (4) That makes sense thank you! Would it then be correct to say that the "p-value" defined here should not be interpreted as a p-value in the classical sense: $P(\delta \ge c|H_0)$ where $H_0$ indicates the treatment has no effect, but rather $P(\delta \ge c|\hat{H}_0)$ where $\hat{H}_0$ indicates the treatment, while running under UniCoRn, has no effect? If so, would the interpretation then be that a p-value < 0.001 means we should be very confident in there being a significant treatment effect, but we should probably not read too deeply into the actual numerical value assigned to it since it might be off by quite a bit?
> > 4. (5) My apologies, my sentence got cut off here. The complete sentence was: I wasn't sure what the takeaway should be for section 4.3, I understand that you're showing such a system can be deployed on a large platform. However, I was left feeling unsure about the interpretation of the results. The delta % numbers are measuring the recommender systems under consideration, but it doesn't tell us much about the A/B testing system. Ideally, I would have liked to have seen delta % with the treatment recommender as the only deployed system compared to when the control recommender was the default. While that isn't a perfect measure because of time-varying effects, it would be very informative since it would tell us whether we were in the right ballpark with the UniCoRn estimate. However, I do understand that there are potentially engineering limitations here, but the authors should at least discuss the reliability/interpretation of the results they've obtained in this section.

---

> > > ### Author Response · Authors · 2021-08-25
> > > **Response to Reviewer neLM (Round 2)**
> > >
> > > Many thanks for helping us strengthen the paper with your constructive comments.
> > >
> > > (1) We will add all the valuable discussions we are having on ATE-inaccuracy. While the monotonicity conditions mentioned in our previous response are sufficient for the optimality of UniCoRn, they might not be necessary. However, at the moment, we don’t have any good ideas on extending the optimality results to a non-monotonic case. Although we cannot guarantee the optimality of UniCoRn in a non-monotonic case, we can provide bounds on the ATE-inaccuracy in terms of Design-inaccuracy for a class of smooth functions. For example, if ATE-inaccuracy = $f$(Design-inaccuracy) for a Lipschitz continuous function $f(.)$ with $f(0) = 0$, then |ATE-inaccuracy| $\leq c$ $\times$ |Design-inaccuracy| where $c$ is the (smallest) Lipschitz constant for $f(.)$. The results follows from the definition of Lipschitz continuity, i.e., $|f(x) - f(y)| \leq c \times |x - y|$ with $x =$ Design-inaccuracy and $y =0$.
> > >
> > > (4) Many thanks for bringing this up. We agree with your $\hat{H}_0$-based interpretation of the p-value. Yes, the treatment-effect estimates can be impacted by a possible bias induced by the UniCoRn-design. We will add these to the paper.
> > >
> > > (5) In the large-scale production system that we implemented UniCoRn in, it was not feasible to ramp a particular treatment (or model) to 100% because of several other ongoing experiments. Additionally, as you pointed out, it wouldn't be an exact A/B because of temporal effects. These factors made a high-fidelity validation of UniCoRn measurement infeasible in our social network experiments. We instead focused on demonstrating the practical feasibility of building such a framework for a large-scale online recommender system. Although the exact measurement fidelity couldn't be validated for the reasons above, we did expect to observe statistically significant results. This is because we observed in a source-side experiment that the viewers tend to send more invitations under the treatment model than the control model, indicating a positive impact of the treatment model on the receivers of those invitations. Note that these source-side measurements can be accurately obtained from a classical A/B testing setup on the viewer-side. We will clarify these in the paper.

---

> > > > ### Comment · Reviewer_neLM · 2021-08-25
> > > > **thank you for your response**
> > > >
> > > > Thank you for the discussion! Your response was very helpful in clarifying any doubts I had, as such I have bumped my score from a 5 to a 7. As long as the current limitations are clearly signposted early in the paper, I think this will be a strong contribution to neurips.

---

### Decision · Program_Chairs · 2021-09-27

**Decision:**

Accept (Poster)

**Comment:**

In this submission, the authors study a very interesting problem, i.e., how to measure from the producer (or seller) side in a two-sided marketplace. This studied task can have great application potential. Due to this, here, I recommend to accept this submission.

However, I do have the same major concern with reviewer neLM, that is, whether the strong assumptions the authors made can be guaranteed. This can hurt the impact of this submission.

Further, this submission also can be improved based on the comments from all the reviewers and the discussion between reviewers and authors. Hope they find these useful, and make this submission a better one.